# Characterization of *Coxiella burnetii* Dugway Strain Host-Pathogen Interactions In Vivo

**DOI:** 10.3390/microorganisms10112261

**Published:** 2022-11-15

**Authors:** Mahelat Tesfamariam, Picabo Binette, Diane Cockrell, Paul A. Beare, Robert A. Heinzen, Carl Shaia, Carrie Mae Long

**Affiliations:** 1Laboratory of Bacteriology, Division of Intramural Research, National Institute of Allergy and Infectious Disease, National Institutes of Health, Hamilton, MT 59840, USA; 2Rocky Mountain Veterinary Branch, Division of Intramural Research, National Institute of Allergy and Infectious Disease, National Institutes of Health, Hamilton, MT 59840, USA

**Keywords:** *Coxiella burnetii*, Q fever, guinea pig, whole-cell vaccine, hypersensitivity, Dugway, avirulence, QVax, bacterial vaccine, virulence

## Abstract

*Coxiella burnetii* is a Gram-negative, intracellular bacterium that causes the zoonosis Q fever. Among the many natural isolates of *C. burnetii* recovered from various sources, the Dugway group exhibits unique genetic characteristics, including the largest *C. burnetii* genomes. These strains were isolated during 1954–1958 from wild rodents from the Utah, USA desert. Despite retaining phase I lipopolysaccharide and the type 4B secretion system, two critical virulence factors, avirulence has been reported in a guinea pig infection model. Using guinea pig models, we evaluated the virulence, whole-cell vaccine (WCV) efficacy, and post-vaccination hypersensitivity (PVH) potential of a representative Dugway strain. Consistent with prior reports, Dugway appeared to be highly attenuated compared to a virulent strain. Indeed, Dugway-infected animals showed similarly low levels of fever, body weight loss, and splenomegaly like Nine Mile II-infected animals. When compared to a human Q fever vaccine, QVax^®^, Dugway WCV exhibited analogous protection against a heterologous Nine Mile I challenge. PVH was investigated in a skin-testing model which revealed significantly decreased maximum erythema in Dugway Δ*dot/icm* WCV-skin-tested animals compared to that of QVax^®^. These data provide insight into this unique bacterial strain and implicate its potential use as a mutated WCV candidate.

## 1. Introduction

*Coxiella burnetii* is a Gram-negative, intracellular bacterium with a near worldwide distribution [1,2]. The zoonosis Q fever is caused by this bacterium, typically following inhalation of infectious particles generated by infected animals. Q fever encompasses a wide spectrum of clinical disease; however, the most common manifestation is a flulike illness known as acute Q fever [3]. Due to the potentially debilitating nature of Q fever, *C. burnetii*’s pronounced environmental stability, and aerosol infection potential, this pathogen is considered a biodefense threat and has been classified as a select agent by the U.S. Centers for Disease Control and Prevention (CDC)-Division of Select Agents and Toxins (DSAT) [4]. *C. burnetii* Nine Mile II (NMII) Clone 4 (RSA439), is exempt from DSAT regulation and may be manipulated at BSL-2 conditions [5]. This clonal strain expresses truncated lipopolysaccharide (LPS) due to a large chromosomal deletion and has been reported to exhibit avirulence in a guinea pig infection model [6]. In contrast, *C. burnetii* NMI is a virulent strain expressing full-length LPS and is commonly used as a positive control for virulence in animal studies. Beyond laboratory-generated strains such as NMII, clinically and environmentally derived *C. burnetii* strains have been isolated from a wide array of organisms including cats, chiggers, cows, dogs, goats, humans, rodents, sheep, and ticks [7,8,9]. These strains exhibit genetic and phenotypic diversity [7,10,11] despite retaining full-length LPS and the type IVB secretion system (T4BSS), two core *C. burnetii* virulence factors.

In 1959, the isolation of several novel “Dugway” *C. burnetii* strains was reported [12]. These strains were isolated from three rodent species (*Peromyscus maniculatus*, *Dipodomys ordii*, and *D. microps*) from the Great Salt Lake Desert, Utah USA during a three-year collection period (1954–1957) [12]. This marked the first report of *C. burnetii* recovery from wild mammals. The unique biologic characteristics of the Dugway strains were reported soon after these strains were isolated. The authors encountered difficulty passaging yolk sac isolates in infected guinea pigs due to apparent avirulence, a hypothesis further bolstered by guinea pig infection studies [13]. Additionally, hamsters appeared to be more susceptible to the Dugway strain infection and pathology than guinea pigs, also developing higher antibody titers following infection. These findings stood in contrast to results obtained for non-rodent derived isolates (e.g., dairy and clinical human isolates). Recent studies confirmed Dugway strain attenuation in guinea pig intraperitoneal [11] and aerosol [10] infection models. Genetic analysis of the Dugway 5J108-111 strain indicated that this strain has the largest genome with the fewest pseudogenes and insertion sequence elements compared to all other *C. burnetii* strains [14]. These data suggest that the Dugway strains supersede other isolates (e.g., NMI, K Q154, and G Q212) in terms of temporal lineage establishment. Dugway strains possess the unique QpDG plasmid [15] and Dugway-specific plasmid-encoded effector proteins have been identified [16]. It remains to be seen whether chromosomal and/or plasmid genomic sequences lie at the root of Dugway strains’ unique behavior. Further, phylogenetic analysis of various *C. burnetii* strains based on *adaA* gene variation suggested that Dugway strains may be the ancestor of all *C. burnetii* strains [17].

Likely related to host adaptation and tropism, these unique genetic and phenotypic features pose Dugway strains as valuable experimental tools and potential vaccine candidates. Avirulence and/or severe attenuation despite retention of primary virulence factors paired with Dugway strains’ unique genomic content all contribute to its intrigue and utility. Dugway strains possess desired qualities for whole-cell vaccines (WCV); however, protective efficacy has not yet been determined. Further, Q fever WCVs are known to cause potentially severe post-vaccination hypersensitivity responses in pre-immune individuals [18], representing a major roadblock for widespread licensing of existing Q fever vaccines (e.g., Q-VAX^®^). Although mechanisms of post-Q fever vaccination delayed-type hypersensitivity (DTH) are being uncovered using novel animal models [19,20], causative antigens have not yet been determined. Accordingly, despite Dugway strain uniqueness, the reactogenicity of Dugway-based WCVs has not been reported. Here, we sought to characterize Dugway-host interactions in vivo using guinea pig models of infection, vaccine challenge, and post-vaccination DTH. Further, these studies were designed to evaluate the feasibility of Dugway-based WCVs as an improved Q fever vaccine. Thus, we created a mutant Dugway ∆*dot/icm* strain, lacking 23 or 26 genes within the *dot/icm* apparatus encoding the TB4SS apparatus. This strategy has been used for *C. burnetii* NMI and conferred attenuation, retained immunogenicity, and potentially reduced post-vaccination DTH magnitude [21].

## 2. Materials and Methods

### 2.1. Coxiella burnetii Strains, Infection Stocks, and Whole-Cell Vaccines (WCV)

*C. burnetii* strains (NMI RSA 439, Dugway 7D 77-80, and Dugway Δ*dot/icm* clone 7) were propagated in acidified citrate cysteine medium-2 or -D (ACCM-2 or ACCM-D) [22] at 37 °C, 2.5% O_2_, and 5% CO_2_ and were stored at −80 °C in a cell-freezing medium (DMEM with 10% fetal bovine serum and 10% dimethyl sulfoxide) until use. *C. burnetii* Dugway Δ*dot/icm* was constructed as previously described for the NMI Δ*dot/icm* strain [21]. Whole-cell vaccine (WCV) stocks, used for vaccination and skin testing, were cultured as infection stocks and were fixed in 4% paraformaldehyde for at least 12 h, washed in sterile PBS, and ultimately resuspended in USP-grade saline prior to being stored at −80 °C. *C. burnetii* infection stock concentrations were quantified using qPCR to enumerate genomic equivalents (GE) [11] while WCV concentrations were determined via direct bacterial count, as previously described [21,23]. Lipopolysaccharide (LPS) from infection stocks was extracted via a modified hot phenol method and visualized by silver stain, as previously described [24] (Appendix A). In accordance with standard operating procedures approved by the Rocky Mountain Laboratories Institutional Biosafety Committee, any manipulations of *C. burnetii* stocks and infected animal tissue were performed in a BSL-3 laboratory.

### 2.2. Guinea Pigs

Four- to six-week-old Hartley guinea pigs were obtained from Charles River, Wilmington, MA, USA (strain code 051) and were acclimated for at least a week prior to experimental manipulation. Female guinea pigs were utilized in these studies to minimize potentially confounding sex-associated factors (e.g., behavior, body weight, hormonal effects) and are used in accordance with historical Q fever virulence studies [10,25]. Animals were housed in individually ventilated plastic cages (Allentown, Allentown, NJ, USA; two animals per cage) with hardwood Sani-chip bedding (PJ Murphy, Montville, NJ, USA). A high-fiber guinea pig diet (Teklad global high-fiber guinea pig diet; Envigo, Indianapolis, IN, USA, cat n. 2041) and chlorinated, reverse osmosis filtered tap water was administered ad libitum. A 12 h light–dark cycle was maintained in animal housing facilities which were kept at 68–72 °F and 40–60% relative humidity with a 50% set point. Six animals per group were utilized for the experiment evaluating virulence, while four animals per group were utilized in vaccine challenge and post-vaccination hypersensitivity experiments. Animals were housed in approved animal biosafety level 3 (ABSL-3) facilities and manipulated under ABSL-3 standard operating procedures approved by the Rocky Mountain Laboratories Institutional Biosafety Committee and an Institutional Animal Care and Use Committee-approved protocol. Animal experiments and procedures were performed in an Association for Assessment and Accreditation of Laboratory Animal Care-accredited NIH/NIAID animal facility.

### 2.3. Infection Model

On the day of infection, animals were placed under isoflurane-induced sedation using an anesthetic vaporizer with activated charcoal absorption filters (VetEquip Inc., cat. N. 901801 and 931401, Livermore, CA, USA) and subcutaneously implanted with an IPTT-300 transponder (BioMedic Data Systems, Seaford, DE, USA) above the shoulder using a large bore needle. Guinea pigs were then infected with 1 mL of 10^6^–10^7^ GE of *C. burnetii* in USP-grade saline via intraperitoneal injection. Negative control animals were mock infected with USP-grade saline. Body weights, body temperatures, and any behavioral/clinical changes were recorded daily at a consistent time for 14 days following infection. Body temperatures were collected using a DAS-8007-P reader (BioMedic Data Systems) and a temperature of ≥39.5 °C was defined as fever [10,26,27]. Fourteen days post-infection, animals were euthanized. Blood and spleens were collected at euthanasia and processed as previously described and bacterial outgrowth from spleen tissues was quantified by TaqMan qPCR (*groel* gene) [21].

### 2.4. WCV Challenge Model

On the day of vaccination, animals were sedated by isoflurane inhalation and implanted with IPTT-300 transponders as described above. Four guinea pigs per group were vaccinated subcutaneously in the upper back with 0.5 mL of USP-grade saline containing 25–2.5 μg of QVax^®^ or paraformaldehyde-fixed *C. burnetii.* Negative control animals were mock vaccinated with USP-grade saline. Body weights, body temperatures, and behavioral/clinical changes were recorded daily following vaccination for a total duration of 28 days. At 28 days post-vaccination, animals were infected with 1 mL of 10^6^ GE *C. burnetii* (NMI) as described above. Upon euthanasia, blood, mesenteric lymph nodes, and spleens were collected and processed as previously described [21].

For mLN and splenic flow cytometric analysis, single-cell suspensions were aliquoted into 96-well U-bottom plates at a density of 1 × 10^6^ cells per well. Cells were washed in a staining buffer (PBS + 1% bovine serum albumin) and stained using a cocktail of antibodies specific for guinea pig cell surface antigens, including B Cells (clone: MsGp10, fluorophore: S/N unconjugated, BioRad, Hercules, CA, USA, cat. N. MCA567) with secondary antibody (anti-mouse IgG1, clone: RMG1-1, fluorophore: AF700, BioLegend, San Diego, CA, USA, cat. N. 406632), CD4 (clone: CT7, fluorophore: RPE, BioRad, cat. n. MCA749PE), and CD8 (clone: CT6, fluorophore: FITC, BioRad, cat. n. MCA752F). Following surface staining, cells were washed in staining buffer and fixed overnight at 4 °C using Cytofix (BD, San Jose, CA, USA, cat. n. 554655). Following fixation, cells were washed in staining buffer and analyzed on a BD FACSymphony flow cytometer using FacsDiva software (BD Biosciences). Data analysis was performed with FlowJo 10.0 software (TreeStar Inc., Ashland, OR, USA). A minimum of 20,000 events were captured for each sample. Single-stained compensation controls and fluorescence minus one staining controls were included to help set gating boundaries.

### 2.5. Post-Vaccination Hypersensitivity Modeling

The guinea pig post-vaccination hypersensitivity model was performed as previously described [21]. Briefly, four guinea pigs per group were infected with 10^6^ GE of NMI or mock infected with saline and monitored for 42 days. Next, animals were sedated by isoflurane inhalation and skin tested with 0.1 mL of 25, 2.5, and 0.25 μg of *C. burnetii* WCV in USP-grade saline via intradermal injection at three separate sites on the shaved back. Negative control animals were mock skin tested with USP-grade saline. Body weights, body temperatures, behavioral/clinical changes, and skin metrics were recorded daily for 21 days post-skin tests. Skin-testing sites were shaved one day prior to intradermal inoculation (“skin testing”) and one day prior to subsequent skin metric measurement. Erythema diameter and induration severity were measured as previously described [21]. Animals were euthanized 21 days following skin testing. Blood, mesenteric lymph nodes, spleens, and skin biopsies were collected for subsequent analysis, as previously described [21].

### 2.6. Histology

Histology was performed as previously described [20]. Briefly, skin biopsies were fixed in 10% Neutral Buffered Formalin for 48 h, placed in tissue cassettes, and processed with a Sakura VIP-6 Tissue Tek (Torrance, CA, USA) on a 12 h automated schedule using a graded series of ethanol, xylene, and PureAffin. Embedded tissues were sectioned at 5 μm, mounted and dried overnight at 42 °C prior to staining with hematoxylin and eosin using established methods. Biopsy specimens were evaluated using an Olympus BX53 microscope (Tokyo, Japan).

### 2.7. Statistical Analysis

Statistical analyses were conducted using GraphPad Prism version 7.0 (GraphPad Software, La Jolla, CA, USA). Statistical evidence for differences in group means was assessed using two-sample Welch t tests, allowing for unequal variances between groups. For each comparison, we computed Wald-type 95% confidence intervals and describe statistical significance with two-sided p-values. We represent *p*-values in equal to or below 0.05 with a single asterisk (*), *p*-values equal to or below 0.01 with a double asterisk (**), and *p*-values equal to or below 0.001 with a triple asterisk (***) unless otherwise indicated. Error bars represent the standard deviation of a group mean.

## 3. Results

### 3.1. C. burnetii Dugway Is Attenuated in an Intraperitoneal Guinea Pig Model of Q Fever

We first evaluated the virulence of Dugway strain 7D 77-80 using a guinea pig model of intraperitoneal infection (Figure 1A). Guinea pigs were injected with 10^6−7^ genome equivalents (GE) of *C. burnetii* or saline. Next, body temperatures and weights were recorded for 14 days, followed by euthanasia. Body temperatures for saline mock-infected animals remained stable along with NMII (10^6^), Dugway (10^6^), and Dugway ∆*dot/icm* infected animals (Figure 1B). Animals infected with 10^7^ NMII and Dugway displayed low-grade, transient fever profiles, in contrast to NMI-infected animals (10^6^) which experienced sustained, robust fever responses. When body temperatures were normalized to the initial readings at day 0, a clear divergence between NMI-infected animals and all other groups emerged (Figure 1C). Body weight change corresponded with body temperature findings, with NMI-infected animals experiencing higher body weight loss than all other groups (Figure 1D). Following the same trend, NMI-infected animals displayed significant splenomegaly compared to saline, NMII, Dugway, and Dugway ∆*dot/icm*-infected animals (Figure 1E). NMII (10^7^), Dugway (10^6/7^), and Dugway ∆*dot/icm* (10^7^)-infected animals exhibited significant splenomegaly compared to saline mock-infected animals. Day 14 splenic bacterial burdens were highest in NMI-infected animals but also present in NMII and Dugway-infected animals (Appendix A). No *C. burnetii* DNA was detected in saline or Dugway ∆*dot/icm*-infected animals.

### 3.2. C. burnetii Dugway Exhibits Heterologous Protection as a Whole-Cell Vaccine (WCV)

Next, we evaluated the efficacy of *C. burnetii* Dugway as a whole-cell vaccine against a challenge with the virulent Nine Mile I (NMI) strain. Guinea pigs were subcutaneously vaccinated with saline, *C. burnetii* WCV, or QVax^®^ in the upper back (Figure 2A).

Body temperature and body weight were taken for 28 days and vaccinations did not induce alterations in body temperature or weight (Appendix A). Guinea pigs were then intraperitoneally challenged with *C. burnetii* NMI (10^6^ GE) and monitored for 14 days before euthanasia. Mock vaccinated and NMI-challenged guinea pigs (Saline:NMI) experienced sustained fever following NMI infection (Figure 2B), serving as a positive infection control. In contrast, Dugway- and QVax^®^-vaccinated animals appeared to be protected from the development of fever apart from a few transient breakthrough animals at the lowest vaccination dose (0.25 µg). When normalized to day 0 starting body temperatures, these trends persisted (Figure 2C). Body weight change was stable in all groups with the exception of mock vaccinated, NMI-infected animals and QVax^®^ (0.25 µg) vaccinated, NMI-infected animals with both groups losing a higher percentage of body weight than others (Figure 2D). Significant splenomegaly was observed in mock vaccinated, NMI-infected animals compared to uninfected animals (Figure 3A). Significant splenomegaly was not observed in any vaccinated and NMI-infected animals compared to uninfected animals. No significant differences in splenic bacterial burden were detected in vaccinated groups compared to unvaccinated, NMI-infected animals (Figure 3B). Although mesenteric lymph node (mLN) cellularity was not altered in Saline:Saline animals compared to vaccinated groups (Appendix A), spleen cellularity was significantly increased in QVax^®^ (25 and 2.5 µg)- and Dugway (2.5 and 0.25 µg)-vaccinated animals compared to Saline:Saline negative controls (Appendix A). Flow cytometry was performed on secondary lymphoid tissues and the gating strategy for CD4^+^/CD8^+^ T cells and B cells is depicted in Appendix A, respectively. dLN B cell frequency was significantly decreased in Dugway (25 µg):NMI animals and appeared to be generally reduced in additional vaccinated groups (Appendix A). CD4^+^ T cell frequency appeared unaltered regardless of treatment (Appendix A). CD8^+^ T cell frequency was significantly increased in unvaccinated, NMI-infected animals compared to unvaccinated, uninfected animals (Appendix A). Splenic flow cytometry revealed significantly decreased B cell frequency in Saline:NMI, QVax^®^ (2.5 and 0.25 µg):NMI and Dugway:NMI groups compared to that of untreated Saline:Saline animals (Appendix A). CD4^+^ T cell frequency was significantly decreased in Saline:NMI, QVax^®^ (0.25 µg):NMI and Dugway:NMI groups compared to that of untreated Saline:Saline animals (Appendix A). No significant alterations in splenic CD8^+^ T cell frequency were observed in treated groups (Appendix A).

### 3.3. Dugway Δdot/icm Strains Exhibit Reduced Post-Vaccination Erythema

A guinea pig post-vaccination hypersensitivity (PVH) model was employed to evaluate the reactogenic potential of various *C. burnetii* WCVs. Guinea pigs were sensitized by intraperitoneal infection using the NMI or Dugway strains (Figure 4A). Body weights and body temperature were measured for 14 days following inoculation and these data indicated a successful infection (Appendix A). Following sensitization, guinea pigs were intradermally injected with saline or *C. burnetii* WCVs on the back at three different doses (25, 2.5, and 0.25 µg) and skin responses were monitored for 21 days prior to euthanasia. Maximum erythema area was significantly increased for all *C. burnetii* sensitized (infected) and skin-tested guinea pigs compared to Saline:Saline mock-treated animals (Figure 4B). Further, compared to unsensitized, skin-tested animals (Saline:NMI) maximum erythema for NMI:QVax^®^, Dugway:Dugway, and NMI:Dugway groups was significantly increased. Compared to highly reactive NMI:QVax^®^ animals, *C. burnetii* ∆*dot/icm* WCV skin-tested animals demonstrated significantly reduced maximal erythematous responses. Erythema kinetics reflect maximal values (Appendix A). Induration was assessed at various time points post-skin testing; early (5 days; Appendix A) and late (20 days; Figure 4C) intragroup responses were similar. Generally, induration severity was aligned among all sensitized and skin-tested groups at the 25 µg dose. At the 2.5 and 0.25 µg doses, the NMI:Dugway ∆*dot/icm* group appeared to experience reduced early and late induration severity. Despite the occurrence of localized PVH responses, body temperature (Appendix A) and body weight (Appendix A) changes due to skin testing were not detected. Significant splenomegaly was detected in NMI:QVax^®^ and NMI:Dugway groups compared to Saline:Saline control animals (Appendix A). Despite this, spleen cellularity (Appendix A) and mesenteric lymph node cellularity (Appendix A) remained unchanged among groups at day 21 post-skin testing.

A standardized histological scoring scheme was applied to guinea pig skin biopsies collected from skin testing sites (Figure 5A). General histological findings associated with sensitization and skin testing occurred in the dermis, hypodermis, and panniculus muscle. Lesions ranged in severity from minimal to severe with abscess formation. Minimal inflammation was characterized by small foci of macrophages, lymphocytes, and sometimes, heterophils within either the dermis, hypodermis, or panniculus muscle. Mildly inflamed foci were increased in size, may occur in more than one tissue layer and contain more macrophages, lymphocytes, and heterophils. Granulomatous to pyogranulomatous inflammation consisted of epithelioid macrophages, multinucleated giant cells, lymphocytes with heterophils which were often degenerative and extended into multiple tissue layers. Moderate to marked pyogranulomatous inflammation consisted of epithelioid macrophages, multinucleated giant cells, lymphocytes, and clusters of degenerate heterophils sometimes associated with necrotic cores. The most severe lesions had a fully developed foci of liquefactive necrosis mixed with degenerative heterophils forming an abscess. There was some blending or overlap of the categories; for instance, a focus of marked pyogranulomatous inflammation was reported just a short distance away from a developing abscess which did not make it into the tissue section. Histological scoring revealed a generally robust inflammatory response in unsensitized Saline:NMI control samples (Figure 5B). Histology score severity was difficult to parse out among groups due to this finding. Notably, NMI:NMI ∆*dot/icm* animals displayed the least robust histological scores at the lowest skin testing dose site (0.25 µg) and histology scores from NMI:Dugway ∆*dot/icm* animals were analogous to Saline:NMI unsensitized control guinea pigs (Figure 5C).

## 4. Discussion

Initial reports regarding Dugway strain characteristics included attenuation or avirulence in a guinea pig model of infection [12] and high infectivity and antibody responsiveness in a hamster model of infection [13]. Compared to high antibody titers induced by virulent strains in hamsters, guinea pigs, and mice, Dugway isolates only induced comparable titers in hamsters. More recently, Dugway strain avirulence in guinea pig infection models has been replicated [10,11]. Building on these studies, we address the potential of Dugway strain virulence, heterologous WCV protective capacity, and post-vaccination reactogenicity. Our guinea pig infection study data indicate that Dugway isolate 7D 77-80 displays similar virulence potential to the exempted NMII strain (RSA 439, clone 4), widely considered to be avirulent or severely attenuated [6,25]. This assessment is based on the effect of infection on body temperature, weight, and changes to the size and histological composition of the spleen. Animals inoculated with Dugway Δ*dot/icm* yielded a similar clinical profile compared to the saline mock-infected control group, apart from slight splenomegaly in animals inoculated with 10^7^ GE. This data complements former reports of Dugway strain attenuation in the guinea pig model [10,11,13] and avirulence in Δ*dot/icm* strains [21]. Considering the physiologic relevance of the guinea pig model with humans in the context of Q fever [28,29] and the lack of any reported human infections involving Dugway strains, they are likely to exhibit attenuation in humans. Other phase I *C. burnetii* isolates exhibit attenuation in guinea pig infection models, including G Q212, Priscilla/MSU Goat Q177, and P Q238 [10,11]. These isolates are derived from human heart valve samples (G Q212 and P Q238) and a goat cotyledon following abortion (Priscilla/MSU goat Q177). Together, this information indicates factors beyond plasmid type and LPS influence *C. burnetii* virulence potential.

Despite an incomplete understanding of protective antigens involved in WCV immunity, the role of full-length LPS/*O*-antigen has been well-established [25]. Given the presence of full-length LPS in Dugway strains, heterologous protection displayed by Dugway WCV was expected. In a direct comparison to QVax^®^, Dugway WCV demonstrated similar efficacy with neither breakthrough fever nor splenomegaly as evidenced at the lowest dose of QVax^®^. Despite clear protective efficacy after challenge infection, demonstrated by a lack of fever, body weight change, and splenomegaly, *C. burnetii* was detectable via qPCR in spleens of all vaccinated animals. Notably, at 14 days post-infection, *C. burnetii* splenic burden appears to be low and difficult to detect, likely due to host clearance. Regardless, sterilizing immunity was not achieved for QVax^®^ or Dugway WCV, although the absence of clinical disease is notable. Further, a significant increase in mLN CD8^+^ T cell frequency in the Saline:NMI group following infection was not reflected in vaccinated, challenged animals. This observation recalls data reported in earlier studies [21] and may indicate a role for cytotoxic CD8^+^ T cells in primary immunity. Here, we present a comprehensive assessment of fever in the guinea pig model in response to QVax^®^ dose escalation in the intraperitoneal guinea pig infection model. This data will likely prove useful for future comparative vaccine studies, as QVax^®^ is considered the gold standard for protection against Q fever.

In a guinea pig PVH model, the Dugway strain appeared to be as reactive as NMI and QVax^®^. Further, regardless of sensitization strain (Dugway or NMI), Dugway skin-tested animals appeared to experience reactogencitiy comparable to NMI skin-tested animals. Histological characterization of skin-testing sites also appeared comparable between strains. This indicates common PVH antigens shared among Dugway and Nine Mile strains. As previously reported [21], Δ*dot/icm* strains appeared to be less reactive based on several experimental endpoints, including erythema and induration. Indeed, Dugway Δ*dot/icm* demonstrated the most promising reduction in reactogenicity. Beyond the potential contribution of the T4BSS, additional antigens remain to be identified. Newly developed murine PVH models may provide utility in further studies [19,20].

The host species from which Dugway strains were isolated from may contribute to their unique characteristics in guinea pig models. Dugway strains were isolated from deer mice (*Peromyscus maniculatus*) and kangaroo rats (*Dipodomys ordii* and *D. microps*). Despite the recent identification of *C. burnetii* DNA in deer mice (Canada) and wild rodents (Spain), further strain characterization was not performed [30,31]. In laboratory settings, deer mice, kangaroo rats, and other wild rodent species were shown to be susceptible to intraperitoneal *C. burnetii* infection, albeit to a lesser degree than guinea pigs [32,33]. Due to the environmental range of *C. burnetii*, a hypothesis exists that describes wild rodents as disease reservoirs with the potential involvement of ticks in the natural lifecycle of the bacterium separate from or associated with the genesis of the livestock lifecycle of infection [9,32,33]. It is tempting to suggest that wild rodent host adaptation may influence *C. burnetii* characteristics such as virulence and behavior in a distinct host, such as a guinea pig or human. Further study is needed to address this hypothesis and the Dugway strain group will likely prove a valuable resource in this effort.

The presented data build upon historic findings relating to unique *C. burnetii* Dugway strains. Our characterization of Dugway host-pathogen interactions in vivo reveals an attenuated strain with vaccine potential. Specifically, the Dugway Δ*dot/icm* strain appears to be a viable WCV candidate, exhibiting significantly reduced reactogenicity. The unique behavior of Dugway isolates paired with the large amount of unique genomic material contained in these isolates raises many important questions. For example, why are Dugway strains attenuated, what do novel genomic regions encode and are they functionally relevant, and does host adaptation play a role in Dugway strain behavior? This manuscript highlights Dugway strain behavior in vivo and provides a framework for future studies to address these inquiries. Indeed, the unique background and phenotype of the Dugway strain group provide a valuable experimental platform for the study *C. burnetii* pathogenesis and mechanisms of virulence.

## Figures and Tables

**Figure 1 microorganisms-10-02261-f001:**
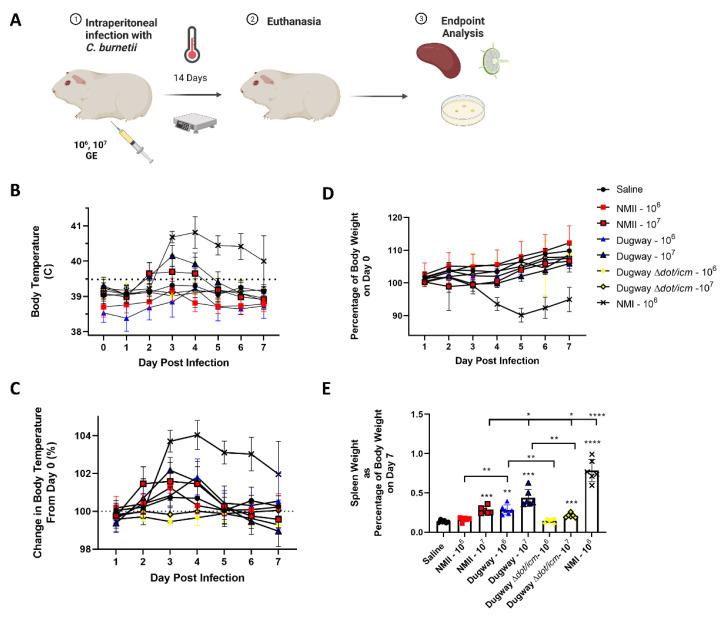
*Coxiella burnetii* Dugway strain displays reduced virulence in a robust guinea pig infection model compared to the Nine Mile strain. The guinea pig model of intraperitoneal infection is outlined in (**A**). Body temperatures (**B**), body temperature change (**C**), and body weight change (**D**) kinetics are displayed for 7 days following infection. Spleens were weighed at euthanasia (day 7) and splenomegaly was determined by normalizing spleen weight to total body weight (**E**). * *p* ≤ 0.05; ** *p* ≤ 0.01; *** *p* ≤ 0.001, **** *p* ≤ 0.0001.

**Figure 2 microorganisms-10-02261-f002:**
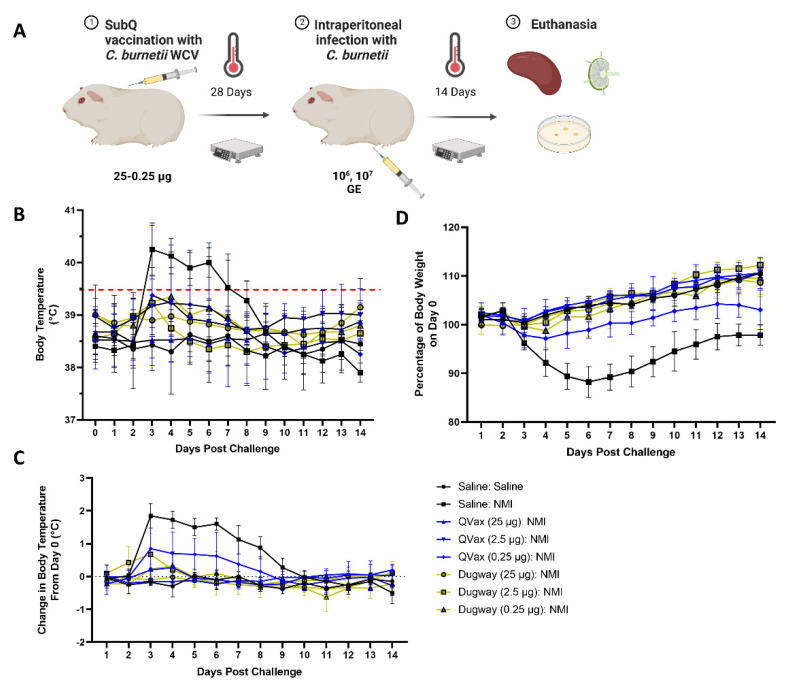
*C. burnetii* Dugway whole-cell vaccine exhibits heterologous protection against fever and body weight loss. The guinea pig vaccine-challenge model is outlined in (**A**). Body temperatures (**B**), body temperature change (**C**), and body weight change (**D**) kinetics are displayed for 14 days following infection.

**Figure 3 microorganisms-10-02261-f003:**
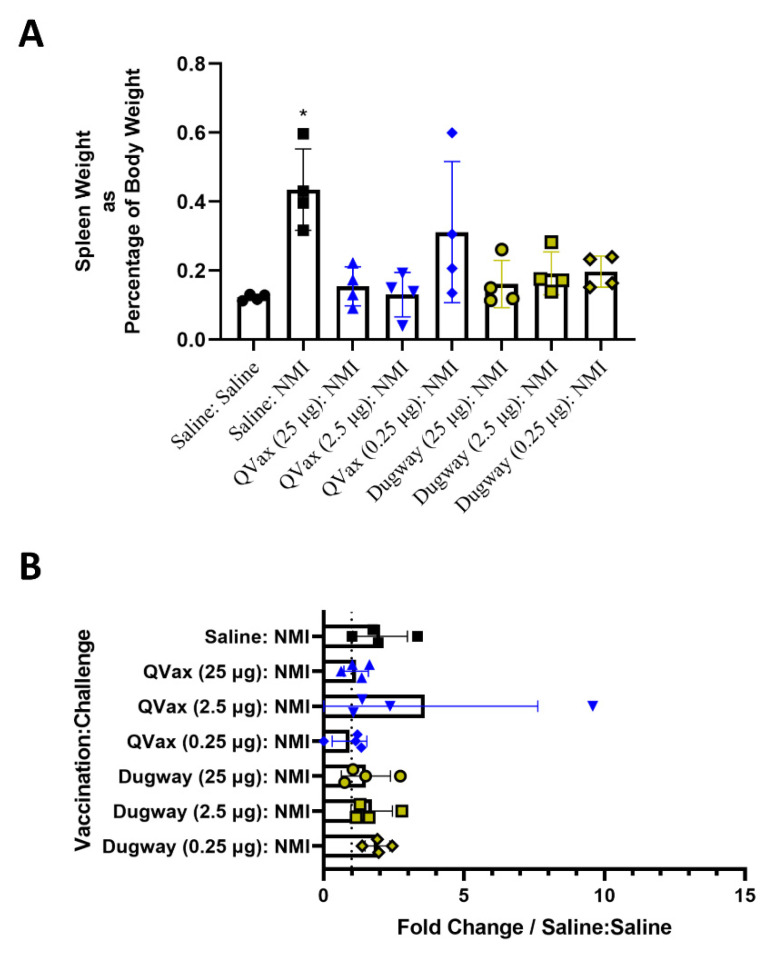
*C. burnetii* Dugway whole-cell vaccine prevents post-infectious splenomegaly but does not alter bacterial burden. Spleens were weighed at euthanasia (day 14) and splenomegaly was determined by normalizing spleen weight to total body weight (**A**). Splenic bacterial burden at euthanasia is represented by fold change in genome equivalents compared to uninfected (Saline:Saline) animals in (**B**). * *p* ≤ 0.05.

**Figure 4 microorganisms-10-02261-f004:**
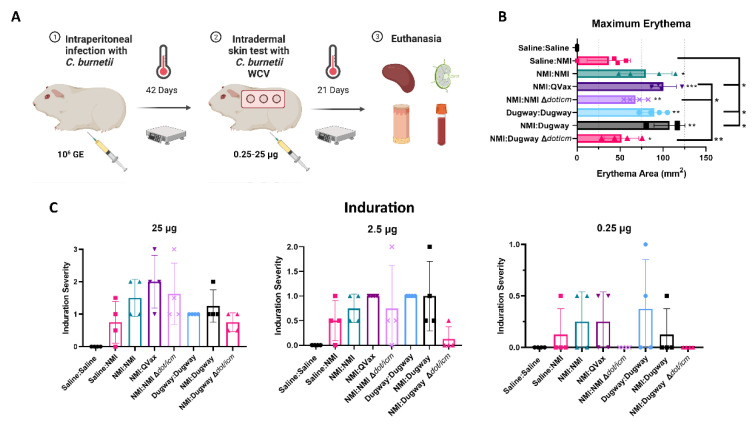
*C. burnetii* Dugway ∆*dot/icm* whole-cell vaccine reduces post-skin testing erythema and late-phase induration magnitude. The guinea pig model of post-vaccination hypersensitivity is outlined in (**A**). Maximum erythema at any given timepoint post-skin testing is depicted in (**B**). Induration scoring values at indicated skin testing site at day 20 post-skin testing (**C**). * *p* ≤ 0.05; ** *p* ≤ 0.01; *** *p* ≤ 0.001.

**Figure 5 microorganisms-10-02261-f005:**
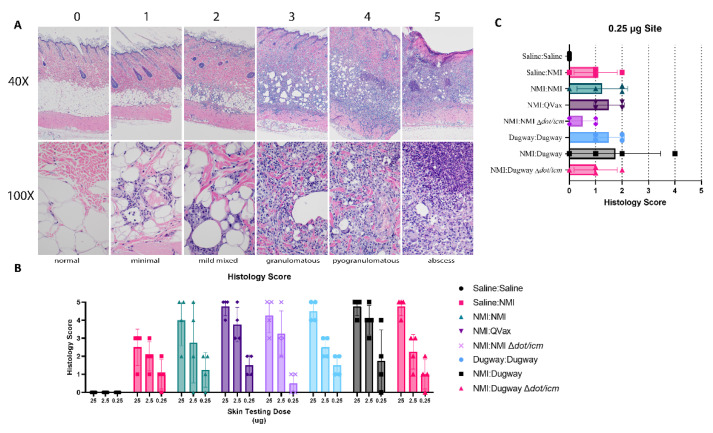
Low-dose skin testing inflammatory responses are reduced following ∆*dot/icm* whole-cell vaccination. Skin-testing site biopsy histological scoring is outlined in (**A**) with inflammatory score (0–5), magnification (40×, 100\×), and brief description denoted. Guinea pigs are identified as follows: 0—Saline:Saline, 25 µg, 1—Saline:NMI, 25 µg, 2—Saline:NMI, 2.5 µg, 3—Saline:NMI, 25 µg, 4—QVax:NMI, 25 µg, 5—QVax:NMI, 25 µg. Inflammatory scoring at euthanasia (day 21 following skin testing) is reported in (**B**). The legend is formatted as infection Strain:Skin testing WCV. Histological scoring at the 0.25 µg site is highlighted in (**C**).

## Data Availability

The original contributions presented in the study are included in the article and Appendix A. Further inquiries can be directed to the corresponding author.

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
