# Peer review of "Characterization of Coxiella burnetii Dugway Strain Host-Pathogen Interactions In Vivo"

_microorganisms, 2022, doi:10.3390/microorganisms10112261_

Round 1

Reviewer 1 Report

This report presents an initial evaluation of a Coxiella burnetii strain (Dugway) with apparent attenuated virulence for potential use in production of an inactivated vaccine. Reactogenicity concerns and the challenges of BSL3 manufacturing have impeded regulatory approval of QVax, the single commercial vaccine for protection against Coxiella-caused Q fever in humans, outside of Australia. Thus, this work will find interest with the occupational health and biodefense communities for which Q fever is of concern. Addressing a number of items prior to publication will enhance the clarity and interest of the report:

(1) The authors report evaluation of the impact of vaccination and infection challenge on major immune cell populations by flow cytometry.  While these assessments address general immune system status in key organs (lymph nodes and spleen), they do not address Coxiella antigen-specific responses.  Were specific humoral responses (circulating antibodies) or cellular responses (e.g., as measured by IFNγ release following ex vivo antigen stimulation of isolated immune cells) evaluated?  If so, or if it were possible to do so with banked study material, it would be good to include these data, and this would strengthen the overall report.  The degree to which antigen-specific responses are or are not observed would be interesting in the context of the reported prevention of symptomatic disease without the conferral of sterilizing immunity (i.e., without preventing bacterial growth following challenge), and may inform selection of appropriate correlates of protection for Q fever vaccine testing in the guinea pig challenge model.

(2) Readability of many of the figures would benefit from enlarging the panels—especially for the various diagrams of study designs (e.g., Fig. 1A) several of which nearly require a microscope to decipher if not enlarged in an electronic reader.

(3) The description of the dose series for the WCV reactogenicity challenge in lines 141-142 is confusing and might more clearly be phrased as “…with 0.5 mL of USP-grade saline containing 2.5-25 ug of QVax or of paraformaldehyde-fixed C. burnetii.”

(4) The legend to Figure 3 does not specify the description of panel B, which presumably should be indicated at the end of line 269.  Also, “fold change in expression” implies measurement of gene expression (RNA or protein), whereas the methods indicate that the measurement was based on qPCR of a genomic DNA preparation, thus “fold change in genome equivalents”.

(5) The legend for Figure 4 appears to be a copy of the Figure 3 legend and should be revised to describe Figure 4.

(6) The legend for Figure 5A indicates that this panel outlines the histological scoring.  However, it only provides images with no indication of which treatment, histological feature, or score is being illustrated by each image.  More information is needed on how the images relate to scoring is needed for full interpretation of Figure 5.

(7) While the manuscript overall is clearly written, the Discussion section will benefit from a careful review for grammatical and other errors.  Among these:

            Line 368: “is recalls” – delete “is”

            Line 370: “QVax’s dose-fever response” is a bit confusing.  Perhaps something along the lines of “…we present a comprehensive assessment of fever in the guinea pig model in response to QVax dose escalation…”?

            Line 371: “QVax is considered the pinnacle of protection against Q fever” implies that no improved protection will be possible.  Is that what the authors wish to suggest?

            Line 383: “The host species that Dugway strains were isolated from…” may more properly be “The host species from which Dugway strains were isolated…”

            Line 397: “The presented data builds…” should be “The presented data build…” (“data” are plural)

            Line 402: English grammar purists will read “begs many important questions” as “avoids many important questions”.  Perhaps “raises many important questions” as an alternative?

Author Response

This report presents an initial evaluation of a Coxiella burnetii strain (Dugway) with apparent attenuated virulence for potential use in production of an inactivated vaccine. Reactogenicity concerns and the challenges of BSL3 manufacturing have impeded regulatory approval of QVax, the single commercial vaccine for protection against Coxiella-caused Q fever in humans, outside of Australia. Thus, this work will find interest with the occupational health and biodefense communities for which Q fever is of concern. Addressing a number of items prior to publication will enhance the clarity and interest of the report:

We appreciate the reviewer’s positive comments regarding the manuscript. We agree that the clarity and interest of the report will be enhanced with the reviewer’s suggestions.

(1) The authors report evaluation of the impact of vaccination and infection challenge on major immune cell populations by flow cytometry.  While these assessments address general immune system status in key organs (lymph nodes and spleen), they do not address Coxiella antigen-specific responses.  Were specific humoral responses (circulating antibodies) or cellular responses (e.g., as measured by IFNγ release following ex vivo antigen stimulation of isolated immune cells) evaluated?  If so, or if it were possible to do so with banked study material, it would be good to include these data, and this would strengthen the overall report.  The degree to which antigen-specific responses are or are not observed would be interesting in the context of the reported prevention of symptomatic disease without the conferral of sterilizing immunity (i.e., without preventing bacterial growth following challenge), and may inform selection of appropriate correlates of protection for Q fever vaccine testing in the guinea pig challenge model.

We agree that Coxiella antigen-specific immune responsiveness would be an interesting approach. We did not measure Coxiella-specific humoral or cellular responses and we do not have banked material to address specific cellular responses. Here, we demonstrated vaccine-induced protective responses that had a measurable influence on disease outcome (e.g. body temperature, body weight, splenomegaly). For the scope of this manuscript, we feel that these outcomes are sufficient to demonstrate vaccine-mediated protection by the Dugway strain.

(2) Readability of many of the figures would benefit from enlarging the panels—especially for the various diagrams of study designs (e.g., Fig. 1A) several of which nearly require a microscope to decipher if not enlarged in an electronic reader.

We agree that the readability of the figures should be enhanced and have enlarged study design images in Figures 1, 2, and 4, as suggested.

(3) The description of the dose series for the WCV reactogenicity challenge in lines 141-142 is confusing and might more clearly be phrased as “…with 0.5 mL of USP-grade saline containing 2.5-25 ug of QVax or of paraformaldehyde-fixed C. burnetii.”

This has been corrected.

(4) The legend to Figure 3 does not specify the description of panel B, which presumably should be indicated at the end of line 269.  Also, “fold change in expression” implies measurement of gene expression (RNA or protein), whereas the methods indicate that the measurement was based on qPCR of a genomic DNA preparation, thus “fold change in genome equivalents”.

We have included the description of Figure 3B in the legend and corrected the wording regarding fold change as directed.

(5) The legend for Figure 4 appears to be a copy of the Figure 3 legend and should be revised to describe Figure 4.

This has been corrected.

(6) The legend for Figure 5A indicates that this panel outlines the histological scoring.  However, it only provides images with no indication of which treatment, histological feature, or score is being illustrated by each image.  More information is needed on how the images relate to scoring is needed for full interpretation of Figure 5.

We added the requested descriptors to Figure 5 and included animal group identity in the figure legend for further clarity.

(7) While the manuscript overall is clearly written, the Discussion section will benefit from a careful review for grammatical and other errors.  Among these:

            Line 368: “is recalls” – delete “is”

This has been corrected.

            Line 370: “QVax’s dose-fever response” is a bit confusing.  Perhaps something along the lines of “…we present a comprehensive assessment of fever in the guinea pig model in response to QVax dose escalation…”?

This has been added.

            Line 371: “QVax is considered the pinnacle of protection against Q fever” implies that no improved protection will be possible.  Is that what the authors wish to suggest?

This is an excellent point and we have replaced “pinnacle” with “gold standard” as not to imply the impossibility of improved protection.

            Line 383: “The host species that Dugway strains were isolated from…” may more properly be “The host species from which Dugway strains were isolated…”

This has been corrected.

            Line 397: “The presented data builds…” should be “The presented data build…” (“data” are plural)

This has been corrected.

            Line 402: English grammar purists will read “begs many important questions” as “avoids many important questions”.  Perhaps “raises many important questions” as an alternative?

This has been corrected as suggested.

Reviewer 2 Report

I thank the authors for this contribution. All in all it´s sound and a well-written composition. I have some minor points to address ro enhance understandability. 

line 19 - 22: Nine Mile I and Nine Mile II are not explained in the introduction. To make the content more understandable it might be helpful to put it like: Dugway infected animals showed similarly low levels of fever, weight loss and splenomegaly like Nine Mile II infected animals. 

line 43 - 44: It makes sense to sort the mammals and insect vectors in order to make it less random. I generally like the inclusion of camels and birds to show the really broad range across the chordata and I would start with the humans.

line 208 significant splenomegaly

line 342: This assessment is based on effect of infection on body temperature, weight and changes to the size and histologic composition of the spleen.

line 349: is this the only reference available? The transferability from guinea pigs to humans is a salient point and needs any confirmation there is.

line 361: despite clear protective efficicacy after challenge infection, demonstrated by a lack of fever....

Author Response

I thank the authors for this contribution. All in all it´s sound and a well-written composition. I have some minor points to address ro enhance understandability. 

 We appreciate the reviewer’s positive comments regarding the manuscript. We agree that the clarity of the report will be enhanced with the reviewer’s suggestions.

line 19 - 22: Nine Mile I and Nine Mile II are not explained in the introduction. To make the content more understandable it might be helpful to put it like: Dugway infected animals showed similarly low levels of fever, weight loss and splenomegaly like Nine Mile II infected animals. 

This has been corrected in the abstract (lines 19-22). We also included more information regarding NMI in the introduction (lines 40-42) for additional clarity.

line 43 - 44: It makes sense to sort the mammals and insect vectors in order to make it less random. I generally like the inclusion of camels and birds to show the really broad range across the chordata and I would start with the humans.

We were unable to apply these suggestions as the mentioned animal species were not utilized in the submitted manuscript.

line 208 significant splenomegaly

This has been corrected.

line 342: This assessment is based on effect of infection on body temperature, weight and changes to the size and histologic composition of the spleen.

 This has been corrected.

line 349: is this the only reference available? The transferability from guinea pigs to humans is a salient point and needs any confirmation there is.

This reference describes human dose-response fever curves in direct comparison to that of guinea pigs via two routes of infection. We feel that this is a strong example of the physiological relevance of the guinea pig model. We included a citation for a recent review article (ref. 29) further addressing this point.

line 361: despite clear protective efficicacy after challenge infection, demonstrated by a lack of fever....

This has been corrected.